# Bioactive Composite for Orbital Floor Repair and Regeneration

**DOI:** 10.3390/ijms231810333

**Published:** 2022-09-07

**Authors:** Fahad AL-Hamoudi, Hamza U. Rehman, Yasir A. Almoshawah, Abdullah C. S. Talari, Aqif A. Chaudhry, Gwendolen C. Reilly, Ihtesham U. Rehman

**Affiliations:** 1Dental Technology Department, King Khalid University, Abha 62529, Saudi Arabia; 2Bioengineering, School of Engineering, Lancaster University, Lancaster LA1 4YW, UK; 3Leeds General Infirmary, Leeds Teaching Hospitals Nhs Trust, Leeds LS1 3EX, UK; 4Mechanical Engineering Department, College of Engineering, Shaqra University, Dawadmi 11911, Saudi Arabia; 5Interdisciplinary Research Center in Biomedical Materials (IRCBM), CUI, Lahore Campus, Lahore 54000, Pakistan; 6INSIGNEO Institute for In Silico Medicine, University of Sheffield, The Pam Liversidge Building, The Sir Robert Hadfield Building, Mappin Street, Sheffield S1 3JD, UK

**Keywords:** polyurethane, hydroxyapatite, repair and regeneration, bioactive composite, angiogenesis, ex-ovo chick chorioallantoic membrane (CAM) assay, blowout fracture

## Abstract

In the maxillofacial area, specifically the orbital floor, injuries can cause bone deformities in the head and face that are difficult to repair or regenerate. Treatment methodologies include use of polymers, metal, ceramics on their own and in combinations mainly for repair purposes, but little attention has been paid to identify suitable materials for orbital floor regeneration. Polyurethane (PU) and hydroxyapatite (HA) micro- or nano- sized with different percentages (25%, 40% & 60%) were used to fabricate bioactive tissue engineering (TE) scaffolds using solvent casting and particulate leaching methods. Mechanical and physical characterisation of TE scaffolds was investigated by tensile tests and SEM respectively. Chemical and structural properties of PU and PU/HA scaffolds were evaluated by infrared (IR) spectroscopy and Surface properties of the bioactive scaffold were analysed using attenuated total reflectance (ATR) sampling accessory coupled with IR. Cell viability, collagen formed, VEGF protein amount and vascularisation of bioactive TE scaffold were studied. IR characterisation confirmed the integration of HA in composite scaffolds, while ATR confirmed the significant amount of HA present at the top surface of the scaffold, which was a primary objective. The SEM images confirmed the pores’ interconnectivity. Increasing the content of HA up to 40% led to an improvement in mechanical properties, and the incorporation of nano-HA was more promising than that of micro-HA. Cell viability assays (using MG63) confirmed biocompatibility and CAM assays confirmed vascularization, demonstrating that HA enhances vascularization. These properties make the resulting biomaterials very useful for orbital floor repair and regeneration.

## 1. Introduction

Orbital floor bone is part of the orbital cavity, which is a highly vascularized tissue. Among all orbital cavity walls, orbital floor blowout fractures are the most common, with a percentage of 50.2% [1,2,3]. The bone fracture is usually a consequence of trauma or disease. The repair and regeneration of orbital floor fractures are challenging, as the disruption of the blood vessel network reduces blood supply to the osteoblasts, significantly impairing healing [3,4,5].

Therefore, there is a need to fabricate bone substitute materials which can help to repair and regenerate bony defects that result due to orbital floor fracture. The aim is to design bioactive scaffolds to correct abnormal bone, assist the healing of fractured bone, and restore missing or damaged bone. A biomimetic bone scaffold material should be osteoconductive, osteoinductive and osteogenic [6] in combination with essential properties such as biocompatibility, bioresorbability and reasonable mechanical strength-mainly compressive strength [7]. Osteoconduction requires enough porosity for cell adhesion and migration, osteoinductive properties promote stem cell differentiation towards osteoblast formation and osteogenesis involves stem cell metabolism, upregulation of growth factors and bone regeneration [8,9,10,11,12].

Orbital floor bone is a natural composite material comprising organic matrices such as collagen fibrils and mineral components, hydroxyapatite being the major component. Synthetic hydroxyapatite (HA] has been widely examined as a repair material, due to its similarity to the main natural component of bone and due to its good osteoconductivity and bioactivity. However, synthetic HA has been used for non load-bearing or low load-bearing applications, due to its limitation in fatigue failure and compressive strength [13,14,15,16,17]. Polyurethane (PU) is a synthetic elastomer with versatile properties which can be tailored to requirements. PUs are a polymer family that contain a urethane linkage in their chain; the main segments are hard and soft. The hard segment is an isocyanate and a chain extender, and the soft segment is a polyol. These segment types and their respective ratios control the properties of the polymer [18]. PU has gained attention for use in clinical application due to its beneficial properties of biocompatibility, outstanding mechanical strength, viscoelastic performance and exceptional elasticity [19]. Even though PU is structurally a promising scaffold material, its bioactivity is limited which restricts its use in bone fractures.

Therefore, the combination of Polyurethane (PU) and hydroxyapatite (HA) may provide a better bioactive composite for biomedical applications, such as orbital floor repair and regeneration. The importance of the composite is the ability to tailor its mechanical and biological properties to the desired application, by altering the amount and type of the reinforcing material and polymer [18,19,20,21]. The bioactive composite properties are affected by the interface between the HA particles and PU. The homogeneous dispersion of the composite leads to an expected improvement in the mechanical properties [22,23]. Nano HA mimics natural bone particles, hence adding nano-HA to a scaffold upregulates osteogenic genes and osteogenic cell attachment to the scaffold [21,24,25,26]. Chemical bonding between the functional groups of HAp and cyano groups of pre-polymer is achieved via covalent bonding, which serves to improve the mechanical properties of the resulting polymer. This has also been reported by Liu et al. [27] and Dong et al. [28] in studies, where they examined the reactivity between isocyanate and HAp and calcium hydrogen phosphates respectively. They concluded that there was a linkage between the HAp and the isocyanate that is created through covalent bonding, whereas a urethane linkage was formed between hexamethylene diisocyanate (HMDI) and calcium hydrogen phosphate. This demonstrates that the OH groups on the surface of the HAp have high reactivity with organic functional groups. Regarding the use of solvent, the research of Gorna and Gogolewski [10] compared scaffold fabrication using solvents such as DMF, DMSO, methyl-2-pyrrolidone (NMP), acetone (A), ethanol (EtOH), isopropanol (I) and tetrahydrofuran (THF). They concluded that DMF is the best solvent for polyurethane scaffold synthesis because it results in open, interconnected pores and higher water permeability.

In this study, the primary aim is to create a biocomposite with improved biocompatibility and bioactivity and mechanical properties of PU/HA scaffolds by determining the most suitable HA concentration to induce vascularisation. The ultimate aim is to design a scaffold suitable for use in orbital floor bone repair and regeneration. A range of ion substituted HAs (Carbonate and fluoride), grafted Hydroxyapatite (Citrate), or co- substituted grafted HAs will be combined with PU scaffolds to ascertain their potential for orbital floor fracture repair and regeneration. The synthesis of HA was reported in our previous study [29].

## 2. Results

### 2.1. PU and PU/HA Scaffolds

All planned scaffolds were successfully fabricated with the exception of PU/HA with 60% loading of micro HA, which was difficult to collect from the petri dish even after 5 days of particulate leaching, where a thin shell of the scaffold’s bottom surface remained on the petri dish. This was not the case with the PU/HA_nano 60%_ scaffold. This could be an initial indication that the HA nano size is better distributed through the scaffold compared to the micro HA. Laser cutting of the scaffolds demonstrated further early indicators about the location and distribution of HA particles. Black burn marks were noted in only the bottom surface of PU/HA_micro_ scaffolds but were found on both surfaces for PU/HA_nano_ scaffolds. This shows that the majority of micro-HA was present in the bottom surface of the scaffolds, but in equal position for the HA nano. 

SEM images were used to obtain measurements of cross-sectional area, porosity, interconnectivity and wall thickness of scaffolds. Figure 1. The well-interconnected porous structure of PU, PU/HA_micro 25%_ and PU/HA_nano 25%_ scaffolds is confirmed by SEM images at different magnifications, Figure 2. Pore interconnectivity, size and morphology play a significant role in scaffold properties. Furthermore, the presence of pores and HA particles on scaffold surfaces were evaluated by increasing the magnification of SEM images. HA particles are present on the scaffold surface of PU/HA_micro or nano_ samples showing that HA acts as a filler in PU. The formed pores’ size in the scaffold were based on the size of salt particles and pore size 450 to 10 μm were obtained (450, 300, 200, 50 and 10 μm).

### 2.2. PU, PU/HA_micro_ and PU/HA_nano_ Scaffolds

FTIR-PAS was used to characterise the PU and PU-HA scaffolds (Figure 3 and Figure 4) and Table 1. For PU scaffolds, the peak at the range 3330 ${\mathrm{cm}}^{-1}$ was attributed to stretching v(N–H). The peaks at 3121 cm^−1^ were the overtone of 1539–1540 ${\mathrm{cm}}^{-1}\;$and 3039 ${\mathrm{cm}}^{-1}\;$attributed to the v(C–H) in a benzene ring. The peaks at 2958 and 2885 ${\mathrm{cm}}^{-1}\;$were CH^2^ peaks of the polyester. The peak at 2958 ${\mathrm{cm}}^{-1\;}$was the asymmetric stretching peak of CH^2^ and the peak at 2885 ${\mathrm{cm}}^{-1}$ was the symmetric stretching of CH_2_. The peak due to bonded C=O stretching was at 1701 ${\mathrm{cm}}^{-1}$ and the free C=O stretching appeared at 1734 ${\mathrm{cm}}^{-1}$. The peak at 1597 ${\mathrm{cm}}^{-1}$ was assigned to v(C=C) in the benzene ring and 1539 ${\mathrm{cm}}^{-1}$ was the amide II δ (N–H)þv(C=N). 1473 ${\mathrm{cm}}^{-1}$ was the weak CH_2_ peak and the 1417 ${\mathrm{cm}}^{-1}\;$attributed to the strong v(C–C) in the benzene ring. The peak at 1311 ${\mathrm{cm}}^{-1}$ was assigned to amide III δ(N–H)þv(C=N), β(C–H) peak and δ(N–H)þv(C=N) appeared at 1232 ${\mathrm{cm}}^{-1}$. The peak at 1081 ${\mathrm{cm}}^{-1}$ was very strong vs. (CH_2_–O–CH_2_) of ester peak. The peak at 1019 ${\mathrm{cm}}^{-1}$ was the weak β(C–H) in the benzene ring and 817 ${\mathrm{cm}}^{-1}$ was the γ(C–H) from butane diol. The peak at the 517 ${\mathrm{cm}}^{-1}$ is attributed to v(C–C) in the benzene ring. These data were similar to Rehman, Khan et al. and Tetteh et al. [19,30,31].

For PU/HA_micro or nano (25%, 40% & 60%)_ scaffolds, as reported by Rehman and Bonfield, the stretching O–H was observed at 3570 ${\mathrm{cm}}^{-1}$ [32]. The bands at 1074, 962, 603 and 566 ${\mathrm{cm}}^{-1}$ were assigned to vibration of the phosphate group, PO_4_. The peak at 1074 ${\mathrm{cm}}^{-1}$ was attributed to the triple degenerated vibration v3, and 962 ${\mathrm{cm}}^{-1}$ was assigned to the non-degenerated symmetric stretching mode v1, of the P–O bond of the phosphate group. The peaks at 603 and 566 cm^−1^ refer to a triple degenerated bending mode v4, of the O–P–O bond. The peak at 635 ${\mathrm{cm}}^{-1}$ was observed due to the phosphate v4 bending. The P–O (stretching and bending) and stretching O–H peaks were not found in the polyurethane spectrum. After introducing the hydroxyapatite into polyurethane, the presence of HA characteristic peaks was recognised. Additionally, there was appearance and shifting of new peaks in the region of 1108–850 ${\mathrm{cm}}^{-1}$. These peaks were attributed to the linkage OP–HC–O (linkage of PO_4_ and vibrational (CH_2_–O–CH_2_). The shoulder peaks at 1108–850 ${\mathrm{cm}}^{-1}$ were evidence of P–O–C formation. This proves obtaining HA in the soft segment of PU which is one of our project objectives [30,33].

FTIR-ATR was used to characterise the PU and PU/HA scaffolds’ top and bottom surface (Figure 5 and Figure 6). The prominent peaks are found, but the significant difference is in the peaks’ intensity, especially the OH, CO_2_ and PO_4_ groups. For the PU scaffold, the spectra of the top and bottom coincided, which means the chemical composition of the top and bottom are the same (Figure 7A). However, Peak intensity with PU/HA_micro_ and PU/HA_nano_ scaffold surfaces shows clear differences, which explains the differences in the physical appearance of those scaffolds. It is worth noting that as the HA percentage increases in PU/HA scaffolds, the peak intensity of OH and PO_4_ increases in conjunction with CO_2_.

For PU/HA_micro_ scaffolds, the intensity of OH and PO_4_ peaks are significantly higher and, the CO_2_ peak is lower in the bottom surface than the top surface, confirming that the majority of HA particles are in the bottom surface (Figure 7B–D). This distribution of HA particles in PU scaffold produces two surfaces, active and nonactive. The (CH^2^–O–CH^2^) of the ester peak is higher in the top surface compared to the bottom one. However, the case is entirely different with PU/HA_nano_ scaffolds (Figure 7E–G). There is no significant difference in peak intensity of OH, PO_4_ and CO_2_ between the top and bottom surfaces of PU/HA_nano_ scaffolds. All peaks are almost entirely matched between both surfaces in those scaffolds, especially with PU/HA_nano 60%_. Those matching of peak intensity could be a sign of dispersed distribution of HA_nano_ particles, due to the high surface area and crystallinity of nanoparticles. This leads to a reduction in the possibility of HA_nano_ particles agglomerating, which provides better stability and combined with PU, generates an equal and better distribution of particles than of microparticles.

### 2.3. Mechanical Properties

Although PU/HA_micro 60%_ scaffolds have significantly the highest Young’s Modulus (1.82 MPa) among all PU/HA_micro_ scaffolds, their yield strength (0.38 MPa) is significantly lower. While PU/HA_micro 40%_ has the highest yield strength (0.79) among all micro groups, although this is not statistically significant Figure 8. PU/HA_micro 40% & 60%_ scaffolds failed completely during the tensile test, indicating they were less able to resist strain compared to PU and PU/HA_micro 25%_. Among this group, the PU and PU/HA_micro 25%_ are suitable to use Figure 8A.

Figure 8B compares PU and PU/HA_nano 25%, 40% & 60%_ under tensile testing. Similar to the PU/HA_micro 25%_ scaffold, the PU/HA_nano 25%_ scaffolds have the lowest Young’s modulus (0.64 MPa) and yield strength (0.62 MPa), even lower than PU scaffolds (0.71 and 0.71 MPa, respectively). Moreover, by increasing HA_nano_ percentage up to 40% and 60%, the Young’ s modulus and yield strength increase rapidly, indicating that the PU/HA_nano 60%_ scaffolds have the highest Young’s modulus (2.75 MPa) and yield strength (1.17 MPa) by a significant margin, but fail more often. All scaffolds (PU, PU/HA_nano 25% and 40%_) were not broken under the tensile test, except for the PU/HA_naon 60%_ scaffolds. PU/HA_nano 40%_ scaffolds are a promising percentage to be selected, due to the increased Young’s modulus and yield strength (0.98 and 1.01 MPa, respectively), as well as an increased ability to bear more strain compared to others. PU/HA_nano 40%_ shows suitable mechanical properties in this group (Figure 8).

Figure 8A shows the tensile test curves of PU and PU/HA_micro 25%_ scaffolds which are similar but confirm that PU/HA_nano 25%_ scaffolds have low mechanical properties. This can be due to the low percentage of HA in HA scaffolds. However, PU and PU/HA_micro or nano 25%_ are not broken during the tensile test. Both the PU/HA_micro or nano 40%_ scaffolds lead to an improvement in mechanical strength. Even though the Young’s modulus of PU/HA_micro 40%_ scaffolds (1.38 MPa) is higher than that of nano (0.98 MPa), PU/HA_nano 40%_ scaffolds have higher yield strength (1.01 MPa) than that with micro (0.79 MPa). During the tensile test, the PU/HA_nano 40%_ survive, indicating a better ability to undergo strain. However, the PU/HA_micro 40%_ scaffolds broke. Both the PU/HA_micro and nano 60%_ scaffolds have high Young’s moduli (1.82 and 2.75 MPa, respectively) but this is significantly higher with nano HA. However, the yield strength with PU/HA_nano 60%_ scaffolds increases (1.17 MPa) compared to PU scaffolds (0.71 MPa), whereas with PU/HA _micro 60%_ scaffolds this is lower (0.38 MPa), see Figure 8.

### 2.4. Cell Viability

The PU and PU/HA _micro_ and PU/HA _nano_ scaffolds were seeded and evaluated using Human Osteosarcoma cell line MG63. As shown in Figure 9, all scaffolds showed an increase in cell metabolic activity, which indicates there is cell growth during the 28 day culture period. MG63 cells cultured on PU scaffolds had the highest metabolic activity on Day 14. This was significantly higher than on PU/HA_micro 25% & 40%_ and all PU/HA_nano_ scaffolds. PU scaffolds supported significantly higher cell viability on day 7 than all PU/HA_nano_. However, on day 28, there is no significant difference with PU/HA_nano 60%_ HA. MG63 cell viability on PU/HA_micro 60%_ scaffolds was the highest amongst the micro group between Day 1, 7 and 28. PU/HA_micro 25%_ scaffolds were the lowest on day 1 and 7, but there was no difference between PU and PU/HA_micro 25% &40%_ scaffolds on day 14 and 28 (Figure 9A). For the HA_nano_ group, the PU/HA_nano 60%_ scaffold was the highest overall, followed by 40% then 25% scaffolds (Figure 9B). Cell viabilities on PU scaffolds were observed to decrease between day 14 and 28, whereas on PU/HA_nano 25% & 40%_ scaffolds there was a decrease between days 7 and 14. This could be because of the multiple washings of scaffolds to undertake the alamar blue assay which may have washed out some cells and caused this decrease.

MG63 cells were seeded on PU/HA_micro_ and PU/HA_nano_ with different percentages (25, 40 and 60%) to examine whether a higher HA content could enhance cell proliferation and increase cell viability. As shown in Figure 9 and Figure 10, the cell viability was improved by increasing HA content for both micro and nano HA. However, the cell viability of PU and PU/HA_micro_ scaffolds was higher than on the PU/HA_nano_ scaffolds over 28 days, but with no significant difference. The only significant differences were observed between the PU/HA_micro 60%_ and PU/HA_nano 25%_ scaffolds, where the 60% had higher cell viability compared to the 25% scaffolds.

### 2.5. Collagen

The ultimate test of a scaffold’s ability to support bone tissue engineering is its ability to support bone-like extracellular matrix deposition via collagen and calcified matrix.

Sirius Red was used to study collagen matrix production on the PU and PU/HA _micro and nano_ scaffolds after 28 days of culture. PU and PU/HA _micro_ scaffolds supported the most collagen deposition within the group, with no significant difference. The PU/HA _nano_ scaffolds supported less collagen deposition than the PU and PU/HA _micro_ scaffolds (Figure 11).

### 2.6. Vascular Endothelial Growth Factor (VEGF) Assay

VEGF protein amounts were measured over 28 days of culture. As shown in Figure 12, the VEGF protein amount was similar between PU, PU/HA_micro_ and PU/HA_nano_ scaffolds with no significant difference. Except on the PU/HA_nano 25%_, which was significantly lower than PU/HA_micro 60%_ and PU/HA_nano 40%_. These last scaffolds were the highest among their group as micro and nano respectively.

### 2.7. Ex-Ovo Chick Chorioallantoic Membrane (CAM) Assay

Angiogenesis evaluation of PU scaffolds with and without hydroxyapatite (40%) was carried out with an ex-ovo (shell-less) CAM assay on chick embryos (Figure 13). After placing samples on CAM at day 7 of the incubation, images of newly formed blood vessels attached to samples were taken at day-10 and day-14. The obtained results revealed the pro-angiogenic response of PU and PU/HA_nano 40%_ leading to micro vascularity at adjacent tissues to the samples. Figure 14 shows a comparison of new blood vessel occurrence with the scaffolds, and Figure 15 shows zoom images of those scaffolds.

## 3. Discussion

Orbital floor bone is a low load-bearing and highly vascularised tissue. Therefore, designing a bone tissue scaffold requires a mechanically compatible material which is capable of undergoing a variety of deformations without rupturing. A polyurethane and hydroxyapatite composite is a good choice for orbital floor because of its beneficial properties of bioactivity, biocompatibility, biodegradability, mechanical flexibility and chemistry. These properties allow PU/HA composites to be tailored for particular applications.

The fabrication of PU and PU/HA scaffolds was carried out using a solvent casting-particles leaching process (PL), due to the ability to control scaffold porosity by determining pore particle amount and size respectively–the polymer amount for this method is low. Yet, the interpore openings and pore shape of scaffolds produced is not controllable.

Naturally porous materials, including tissues, typically have a gradient porous structure (GPS), in which porosity is not uniform. Gradient porosity is observed in bone tissues and this optimises the material’s response to external loading. Gradient porosity also enables specific cell migration during tissue regeneration. It is also required for the treatment of articular cartilage defects in osteochondral tissue engineering. For bone tissue engineering, the optimal pore size for osteoblast activity in tissue engineered scaffolds is still controversial as there have been conflicting reports. High porosity is associated with low mechanical strength, so an optimal level of porosity is required to facilitate cell growth and sufficient mechanical strength [11,34] The average human osteon pore size is 223 μm. The ideal optimum pore size reported for scaffolds is between 100 and 350 μm. This facilitates significant bone ingrowth, vascularisation and nutrient delivery for bone tissue engineering applications [35]. Furthermore, it is reported that a combination of different pore size performs better in scaffolds than only one pore size [36]. The scaffolds fabricated here fulfil this requirement as they have a pore-size range from 450 to 10 μm, similar to the results seen in the Gorna and Gogolewski study [10].

PU/HA_nano_ had more porosity than PU/HA_micro_, likely due to the nanoparticles size of HA, nano-sized particles would be better distributed inside PU and this would support a better interaction between PU and HA_nano_ because of the high surface area and crystallinity of nanoparticles compared to micro.

The good distribution and interaction of nano HAinside PU can be evaluated by comparing the peaks’ intensity difference between PU/HA_micro_ and PU/HA_nano_ scaffolds (Figure 5, Figure 6 and Figure 7). For PU/HA_micro_ scaffolds, the intensity of ${\mathrm{OH}\;\mathrm{and}\;\mathrm{PO}}_{4}$ peaks are higher, and the ${\mathrm{CO}}_{2}$ peak is lower on the bottom surface than the top surface, confirming that the majority of HA particles are in the bottom surface (Figure 7B–D). However, the case is entirely different with PU/HA_nano_ scaffolds (Figure 7E–G). There is no significant difference in peaks intensity of $\mathrm{OH},{\;\mathrm{PO}}_{4}$ and ${\mathrm{CO}}_{2}$ between PU/HA_nano_ scaffolds top and bottom surfaces. All peaks are almost matched between both surfaces in these scaffolds, especially with PU/HA_nano 60%_. This matching of peak intensity provides evidence of dispersed distribution of HA_nano_ particles. This leads to a reduced possibility of HA_nano_ particles agglomerating, leading to improved stability. This is similar to Yang et al. and Tetteh et al. studies where the PU scaffold with HA nano had a good distribution [31,37]. It has been reported that nano HA particles improve adhesion between the nanoparticles and the polymer matrix compared to micro HA [38].

The prominent peaks are found, but the significant difference is in the peak intensity, especially the OH, ${\mathrm{CO}}_{2}$ and ${\mathrm{PO}}_{4}$ groups. For PU scaffold, the spectra of the top and bottom are coincidental, which means the chemical composition of the top and bottom are the same (Figure 7A). However, Peak intensity with PU/HA_micro_ and PU/HA_nano_ scaffold surfaces shows a clear difference, which explains the differences in the physical appearances of these scaffolds. It is worth noting that, as the HA percentage increases in PU/HA scaffolds, the peak intensity of OH and ${\mathrm{PO}}_{4}$ increases while it is with$\;{\mathrm{CO}}_{2}$. These findings confirmed the findings of Wang et al., study, as they fabricated PU/HA scaffolds with different concentrations of HA [39]. The FTIR spectra also showed that the peak intensity increased with an increase in the amount of HA [19,31,39,40].

Overall, porosity is also affected by the incorporation of HA, as increasing HA percentage may lead to a decrease in the porosity of scaffolds. This indicates, as confirmed by Liu et al. study, that the porosity of PU/HA_micro (30%, 40% & 50%)_ scaffolds are higher than those with 60% HA [41]. The scaffold architecture and 3D form might affect the functionality of bone constructs and perform a critical purpose in bone formation [42].

In general, scaffolds should have adequate mechanical strength to sustain integrity until new tissue regeneration. Scaffolds fabricated for bone repair and regeneration should have comparable strength to native bone, which in this study is the orbital floor, to resist physiological loadings and to prevent stress shielding from occurring. The mechanical properties of scaffolds can be affected by scaffold porosity and pore size. Mechanically, scaffolds with HA_nano_ can bear more strain than HA_micro_, especially 40% nano HA Figure 8). Ceramic materials are usually brittle, but the nano size has more elasticity compared to others. Nanomaterial interfaces lead to an irregular arrangement of atoms, which makes it easy for the atoms to migrate under the external deformation force. Therefore, incorporation of nano ceramic materials improves the mechanical properties of the materials [23,43,44]. This could be the reason why PU/HA_nano 25%_ has a lower Young’s modulus than PU. Overall, scaffolds that have HA_micro or nano 40% or 60%_ see an increase in their mechanical properties. However, 40% HA_nano_ shows a relatively better ability to strengthen the scaffolds. Similarly, Liu et al. compared mechanical strength of scaffolds with 30%, 40% and 50% micro HA, and they found that the compressive strength of scaffolds increased with the HA content [41]. The current research also supports the finding that the mechanical strength of scaffolds increases as the percentage of HA increases, specifically with nano size.

Cell behaviour is directly affected by the scaffold architecture since the extracellular matrix (ECM) provides cues that influence the specific integrin ligand interactions between cells and the surrounding tissues. Hence, the 3D scaffold environment can influence cell proliferation or direct cell differentiation. The role of porosity and interconnectivity in scaffolds is also to facilitate cell migration within the porous structure such that cell growth is enabled while overcrowding is avoided [11,34]. However, it should be noted that cell differentiation is also dependent on the cell type, scaffold material and fabrication conditions. The chemical composition of inorganic constituent in natural bone has a similarity to hydroxyapatite (Ca_10_(PO_4_)_6_(OH)_2_,HA) [32]. HA has been widely used as a biocompatible material in many areas of medicine. HA is well established as a synthetic material for bone replacement due to its chemical resemblance to the inorganic component of bone and tooth [29]. Additionally, HA is known to promote faster bone regeneration and direct bonding to the regenerated bone without intermediate connective tissue [45]. MG63 cells were seeded on PU/HA_micro_ and PU/HA_nano_ with different percentages (25, 40 and 60%) to examine whether a higher HA content could enhance cell proliferation and increase cell viability. As shown in Figure 10 the cell viability was improved by increasing HA content for both micro and nano HA. This may be because the HA particles can attract more serum proteins around the polymer [46]. Besides, it was proven that the HA particles could enhance the content of oxygen around the surface, and this led to better cell attachment [47]. This has been shown in many previous studies, which developed scaffolds with more than 40 wt% HA [37,48,49]. However, the cell viability of PU and PU/HA_micro_ scaffolds were higher than the PU/HA_nano_ scaffold over 28 days, but with no significant difference. The only significant differences were observed between the PU/HA_micro 60%_ and PU/HA_nano 25%_ scaffolds, where the 60% had higher cell viability compared to the 25% scaffolds. Similar to Popescu et al. study, the PU/HA_nano_ did not improve cell viability compared to PU scaffolds [50]. This may be because of the bioactive properties of HA nano particles with the higher surface area and crystallinity. The high surface area of HA nano facilitates a strong interaction between the polymer and ceramic phase, where the HA particles were encapsulated in the PU matrix and were not exposed enough to cells to be interacting [51,52,53,54,55,56]. However, it should be considered that different cell types may have different responses to hydroxyapatite size and composition. For example, Tetteh et al. found that the cell proliferation of MLO-A5 cells was faster on PU/HA_nano_ scaffolds than PU/HA_micro_ ones, but there was no difference in the proliferation rates of hES-MP cell [31]. Collagen matrix production by cells on PU and PU/HA_micro and nano_ scaffolds on day 28 of culture confirmed the results of cell viability, as the PU/HA_nano_ scaffolds supported significantly less collagen than PU and PU/HA_micro_ (Figure 11B). This could be related to the overall amount of cells on the scaffolds, similar to Tetteh et al. where the scaffolds that supported high cell viability had high collagen [31]. Furthermore, angiogenesis is necessary for not only for bone formation and but also during healing/bone remodelling that shows the importance of angiogenesis for osteogenesis. Recently in our group, heparin has been used to induce angiogenesis. It has been investigated via CAM assay to evaluate the potential attachment of physiologically available angiogenic growth factors to pro-angiogenic receptors by using heparin bonded chemically crosslinked chitosan poly-vinyl alcohol (PVA) hydrogels. Triethyl orthoformate (TEOF) crosslinked and heparin bonded hydrogels led to more blood vessel generation as compared to heparin-free control samples, which promote bone remodelling [57]. In addition, direct mixing of heparin in chitosan-PVA-PCL hydrogels in the absence of any growth factors has been investigated for angiogenesis on chick embryo’s CAM tissues for wound healing application. It was found that heparin bonded chitosan-PVA-PCL hydrogels led to significantly more angiogenesis than the sole collagen control gels [58]. Therefore, the functionalisation of scaffolds for its potential physiological binding activity can provide advantages in bone regeneration and remodelling.

Vascularisation of newly bioengineered tissue is critical for osteointegration, and VEGF is determinant vascularisation potential. Several approaches have been proposed to improve the angiogenic potential of the scaffolds. VEGF releasing scaffolds can enhance neovascularisation and bone regeneration, once implanted [59,60]. These factors diffuse into the local environment, where they induce existing blood vessels to grow into the scaffold, ultimately forming continuous vessels [61]. VEGF protein amounts were measured over 28 days of culture. As shown in Figure 12, the VEGF protein amount was similar between cells grown on PU, PU/HA_micro_ and PU/HA_nano_ scaffolds with no significant difference. However, cells on PU/HA_nano 25%_, scaffolds secreted significantly lower than PU/HA_micro 60%_ and PU/HA_nano 40%_. There is a different reading between the alamar blue cell viability and Sirius Red collagen amount as one group and VEGF amount as another group, especially on the PU/HA_nano_ group specifically with 40%. PU/HA_nano 40%_ showed similar VEGF protein amount to PU/HA micro, wherein cell viability and collagen amount PU/HA_micro 60%_ and micro group were higher. This could be due to the multiple washings of scaffolds during the alamar blue and Sirius Red assays which may have washed out some cells. Whereas there is no washing process for the VEGF assay. The VEGF assay results indicate the high possibility of forming new blood vessels. In the current study, this would be the PU/HA_micro 60%_ and PU/HA_nano 40%_. In combination with these and the mechanical property results PU/HA_nano 40%_ was chosen to take forward for further study. Angiogenesis is the growth of new blood vessels from existing vasculature, which is essential to supply oxygen and nutrients to developing tissues, and to facilitate wound healing. The required minimum porosity for blood vessel regeneration is roughly 30 to 40 mm to enable the exchange of metabolic components. CAM assay of PU scaffolds with and without hydroxyapatite (40%) was carried out. The obtained resultsrevealed the pro-angiogenic response to PU and PU/HA_nano_ (40%) leading to micro vascularity at adjacent tissues to the samples. Although the vascular index of the PU scaffold with 40% of HA_nano_ was slightly higher than PU scaffolds, the difference was not significant (Figure 14 and Figure 15). Additionally, minimal studies are investigating the effect of different concentrations of HA on angiogenesis. In a study by Kocak et al., they evaluated the effect of heparin on the angiogenesis of chitosan/hydroxyapatite (Cs/HA) composite hydrogel gel. They found that with or without heparin, the Cs/HA composite exhibited a proangiogenic response [21,23]. This indicated that the presence of HA can promote a proangiogenic response.

The different concentrations of HA in compositions are under further investigations in our lab. Overall, in terms of mechanical properties, cell viability, collage formation, VEGF protein amount and the pro-angiogenic response of most PU/HA, PU/HA_nano 40%_ could have potential applications as minimally-invasive biomaterials to promote vascularized bone tissue regeneration for orbital floor regeneration.

## 4. Materials and Methods

### 4.1. Materials

Polyurethane pellets Avalon85AB were purchased from Huntsman Holland BV, Rotterdam, The Netherlands. Dimethylformamide (DMF) was obtained from Fisher Scientific, Loughborough, UK. P218R micro-Hydroxyapatite was obtained from Plasma Biotal Limited, Buxton, UK. The nano- Hydroxyapatite were obtained from Sigma Aldrich, Gillingham, UK.

### 4.2. PU Solution

PU solution was prepared by dissolving 10 wt% Avalon85AB PU pellets in Dimethylformamide (DMF) solvent. The solution was stirred in a glass flask with magnetic stirrers for 24 h at room temperature.

### 4.3. PU/HA Solution

PU solution was prepared by dissolving 10 wt% Avalon85AB PU pellets in Dimethylformamide (DMF) solvent. The solution was stirred in glass flask with magnetic stirrers for 24 h at room temperature. After that, micro and nano HA powders were introduced to the PU solution in a weight ratio of 3:1 PU to HA (25% HA), 2.4:1.6 PU to HA (40% HA) and (1.6:2.4) PU to HA (60% HA). The PU/HA solution was stirred for a further 24 h at room temperature. (Table 2).

### 4.4. Preparation of Salt Particles

The salt particles within the range of 450 to 10 μm were selected (being 450, 300, 200, 50 and 10 μm). The required salt amount for fabricating the scaffolds was divided equally between those sizes.

### 4.5. Cast Solvent/Particles Leaching PU and PU/HA Scaffolds

Porous scaffolds were prepared using salt, where the PU/Salt (*v*/*w*) ratio was 0.6 mL/1 g. The salt was physically mixed using a metallic spatula with either PU or PU/HA solution for 5 min in a glass flask. The glass petri dishes were placed in an oven at 37 °C for 96 h for solvent evaporation. To identify whether evaporation had completed, films were checked every 24 h. After 96 h, the films were peeled from the glass petri dishes. The scaffolds were soaked in deionised water with 10% ethanol to leach salt particles on a rotating spinner for 3 days to create the porous form (water was replaced every 24 h). The scaffolds were dried using a desiccator for 24 h. Scaffolds were stored in petri dishes and subjected to further characterisation (Figure 16).

### 4.6. Peaparation of Scaffolds for Charactrisation

The obtained PU and PU/HA scaffolds were cut by EPILOG LASER mini. The required sample shape was designed by CorelDRAW X5 software. For SEM, CT and FTIR characterisation, the required form was 3 mm thickness and 10 mm in diameter. For cell culture, the required form was 3 mm thickness and 5 × 5 mm in diameter. For mechanical tensile tests, the required shape was 3 mm thickness and 3 × 15 mm in diameter. The laser was set in vector mode with 5000 Hz frequency, 10% power and 60% speed. For PU scaffolds, cutting was repeated for 5 times. For 25%, 40% and 60% HA in scaffolds, the repeated cutting times were 10, 15, 30 respectively. The reason for repeating the cut and not increasing the power is to complete the cutting without burning scaffold edges.

### 4.7. Scanning Electron Microscopy (SEM)

TESCAN Vega3 LMU variable pressure SEM was used in this study to observe the porous structure of the PU and PU/HA scaffolds. Samples in the shape of a disk with a diameter of 10 mm were coated with gold using Gold Coater (Quorum SC 7620) Sputtering Time for 1.5 s time.

### 4.8. Fourier Transform Infrared Spectroscopy (FTIR)

Nicolet IS50R FTIR spectrometer (Thermo Nicolet, UK), in conjunction with either an Attenuated Total Reflectance (ATR) or a photoacoustic sampling (PAS) accessory (MTech PAS cell), was used in this study. ATR was used for the initial and surface characterisation, where PSA was used to characterise the bulk, as it permitted analysis of samples without sample preparation (e.g., grinding with KBr and pressing into a transparent disc). The sample chamber of the PAS cell was purged with helium gas. The OMNIC 9™ software was used to obtain and process the ATR and PAS spectral data. The background spectrum was collected using a carbon black film at a resolution of 4 ${\mathrm{cm}}^{-1}$,and mid-IR range at (4000 to 400 ${\mathrm{cm}}^{-1}$) for both ATR and PAS. However, the scanning average was 60 scans for ATR, and 128 scans for PAS. At the same setup, the sample spectrums were collected.

### 4.9. Mechanical Test: Tensile Test

The mechanical properties of the scaffolds were analysed on a tensile testing machine (Electro Force 3200, Bose, Framingham, MA, USA) in tension. Rectangular samples were produced with average dimensions 3 mm × 5 mm × 15 mm and mounted between two grips at a minimum gap length of 6 mm. The samples were subjected to tensile strain at the speed of 6 mm/min. The deformation of samples was detected by the movement of the top grip and the load data was detected by 22N load cell the resultant load/deformation curves were translated into stress/strain curves using the cross-sectional area of samples (n = 6).

### 4.10. Culture Conditions for Human Osteoblast Cell Line MG63

The human osteoblast cell line MG63 was used for assessment of cell attachment and cell viability. Cells were maintained at 37 °C and 5% CO_2_ with Dulbecco’s modified eagle’s medium (DMEM) supplemented with 10% FBS (Advanced protein products, Brierley Hill, UK), 100 IU/mL penicillin, 100 μg/mL streptomycin, 2 mM glutamine, and 0.25% fungizone (Gibco Invitrogen, Paisley, UK). Cells were used for experiments between passages 66 and 72. Experiments were run in triplicates with three samples each.

### 4.11. Prepare and Sterilise Scaffolds

PU and PU/HA_micro and nano_ (25%, 40% and 60%) scaffolds were prepared with 3 mm thickness and 5 × 5 mm diameter. All scaffolds were immersed in 70% Ethanol, covered and then subjected to orbital shaking for 2 h. The ethanol was eliminated and discarded and scaffolds were left to dry for 24 h in a tissue culture cabinet. Scaffolds were immersed in PBS, covered and orbitally shaken for 2 h, with the PBS subsequently eliminated and discarded. Scaffolds were immersed in DMEM medium, covered and shaken orbitally for 10 min. DMEM medium was eliminated and discarded. Scaffolds were securely sealed on 96 well plate and kept in the refrigerator for 24 h before seeding the cells.

### 4.12. Cell Seeding

On the seeding day, PU and PU/HA_micro and nano_ (25%, 40% and 60%) scaffolds were incubated at 37 °C in a humidified environment supplied 5% CO_2_ for 1 h prior to the experiment. Each scaffold (3 mm thickness and 5 × 5 mm in diameter) was seeded with 1 × 10^5^ cell/10 μL (1 × 10^7^ cell/mL) of MG63s and incubated for 4 h to allow cells to attach. 260 μL DMEM medium were added to each scaffold and incubated for 24 h.

### 4.13. Alamar Blue Cell Viability

The resazurin-based Alamar Blue^®^ assay is a method of measuring cell metabolic activity. (AbD Serotec, Kiddlington, UK). During the 28 day cell culture period, the Alamar blue assay was carried out on days 1, 7, 14 and 28. The culture media was removed from the scaffolds and washed with PBS twice. 2 mL of Alamar blue solution at 10 μg/mL (in PBS) was added to each scaffold under light sensitive conditions, foil covered and incubated for 2 h at 37 °C in a humidified environment supplied 5% CO_2_. After incubation Alamar blue fluorescence was read at 540 nm in a plate reader. For ongoing cell culture scaffolds were washed in PBS and then fresh osteogenic working media added. At day 28, samples were washed to remove the AlamarBlue^®^ and cell were fixed with 3.7% formaldehyde.

Absorbance at λ570 nm was measured in a colorimetric plate reader (Bio-TEK, NorthStar scientific Ltd., Leeds, UK). Cell-free scaffolds in DMEM were included as controls. Sirius Red Collagen Staining

Sirius Red staining is a method for determining the presence of collagen (Junqueira et al., 1979). The Sirius Red solution was prepared using Direct red dye from (Sigma-Aldrich, UK) at 1 mg/mL in saturated picric acid. 3–6 mL Sirius Red solution (depending on the size of the scaffolds) was added to each well and left on the orbital shaker for 18 h. After staining, excess Sirius Red solution was removed under running tap water until the solution turned clear. Samples were allowed to air dry for 4 h and photographic images were taken for qualitative analysis. To destain Sirius red 0.2 M of NaOH and methanol at 1:1. 2–4 mL was applied to the scaffolds on an orbital rocker at 30 rpm for 24 h. The absorbance of the eluate was read with a spectrophotometer at 490 nm.

### 4.14. Vascular Endothelial Growth Factor (VEGF) Assay

Vascular endothelial growth factor (VEGF) is a protein which is produced by cells to stimulate the formation of blood vessels. To evaluate the VEGF release from the cells, supernatant was collected on day 28 and stored at −20 °C till used. The Quantine Elisa-VEGF Assay kit was used. Reagents were prepared according to the manufacturer’s instructions. Wash buffer was prepared by mixing 20 mL wash buffer and 480 mL of deionised water and stored at −4 °C.

50 μL of diluent assay RD1W was added to all wells, followed by addition of 200 μL of standard and sample solution to the wells. The wells were covered and incubated at room temperature at 20 rpm on an orbital shaker for 2 h. The wells were aspirated after 24 h, and 300 μL wash buffer used to wash wells-this process was repeated 3 times. 200 μL of human VEGF conjugate was added to all wells, the wells were covered and incubated at room temperature at 20 rpm on an orbital shaker for 2 h. The washing process was repeated. 200 μL of substrate solution was added to all wells, and the plate was covered with aluminium foil and incubated at room temperature on an orbital shaker for 2 h. This was followed by adding 50 μL of stop solution into each well. The well content was transferred into 96 well plates to be read at 490 nm within 30 min.

### 4.15. Chick Chorioallontoic Membrane Assay (CAM Assay)

In this experiment, the process of Eke et al. (2017) was followed [63]. Pathogen-free fertilised white leghorn chicken eggs were obtained from Henry Stewart Co. Ltd. (Fakenham, UK). The experiment was consistent with the guidelines of the Home Office, UK. Chick embryos were cultured in an incubator for up to 3 days. The eggshells were cracked, and embryos were transferred into a square petri dish on embryonic development day (EDD) 3. The ex ovo cultures were maintained in a humidified incubator at 38 °C between EDDs 3 to 14. The survival of the embryos was checked daily and recorded. Materials were disposed of via the animal anatomical waste route.

### 4.16. Scaffold Sterilising and Implanting

PU and PU/HA _nano_ (25%) scaffolds were prepared to a 3 mm thickness and 5 × 5 mm diameter using a laser cutter and sterilised as previously described. On the day of implantation PU and PU/HA _nano_ (25%) scaffolds were incubated at 37 °C in a humidified environment and supplied with 5% CO_2_ for 1 h prior to the experiment. On EDD 8, a scaffold of each group was placed on the CAM surface between the significant blood vessels. On EDD 10 and 14, pictures of scaffolds and surrounding CAM area were taken with a digital microscope. After imaging, embryos were sacrificed by cutting their vitelline arteries, and scaffolds were resected from the CAM surface with a 1 cm margin.

### 4.17. Statistical Analysis

All experiments were completed in triplicate and repeated twice. All data was reported as mean ± standard deviation. Comparison of sample means of a mechanical test, Alamar blue cell viability, collagen and hVEGF absorbance data analysis were performed by one-way analysis of variance using GraphPad Prism. Differences between two groups were distinguished as statistically significant if *p* ≤ 0.05 (*), *p* ≤ 0.01 (**), *p* ≤ 0.001 (***) and *p* ≤ 0.0001 (****) as determined by Tukey’s post-hoc pairwise comparison. For imaging of samples (SEM and CT) and CAM assay, some experiments were limited to one sample per condition, per experiment.

## 5. Conclusions

Bioactive PU/HA scaffolds, containing either micro or nano particles size, were prepared via the particle leaching technique. Different percentages of HA (25%, 40% and 60%) were used to evaluate the most suitable percentage for orbital floor repair and regeneration. Interconnected and porous morphology of scaffolds ranging in pore size from 10 m to 450 m support cell bone cell activity and angiogenesis for bone regeneration. HA nano particles were equally distributed on the PU scaffold surfaces compared to HA micro.

The highest concentration of HA micro or nano (60%) in PU ((1.6:2.4) PU to HA) scaffolds led to an increase in Young’ s modulus, the yield strength increased with the nano size. However, both scaffolds were broken during the tensile test. Besides, the cell viability and collagen formation via these samples was higher than the other percentages of HA within PU scaffolds. The VEGF protein, which is the initial sign of vascularisation, by PU/HA_micro 60%_ was higher than other groups, but similar to PU/HA_nano 40%_.

This finding was confirmed via CAM assay; vascular index obtained via PU/HA _nano 40%_ was higher than those of PU scaffold. In addition, the PU/HA_nano 40%_ scaffold might be the most suitable for orbital floor, due to the mechanical properties as well as the results of cell viability and collagen deposition.

## Figures and Tables

**Figure 1 ijms-23-10333-f001:**
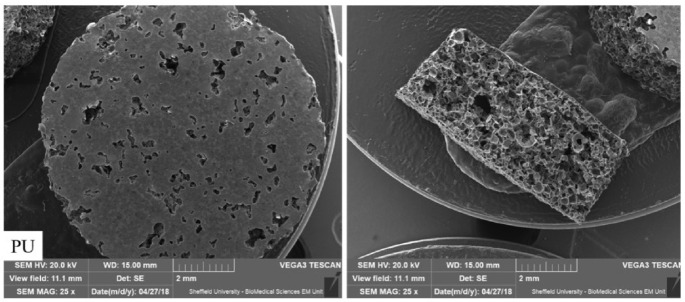
SEM images of PU, PU/HA_micro_ and PU/HA_nano_ scaffolds.

**Figure 2 ijms-23-10333-f002:**
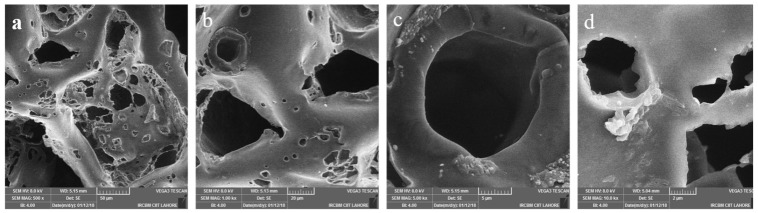
SEM images of PU scaffolds (**a**–**d**) and PU/HA scaffolds (**e**–**h**) at magnification (500×, 1 kx, 5 kx and 10 kx), indicating morphology, interconnectivity and dispersion of HA surface particles. In PU scaffolds no agglomeration of inorganic HA can be observed as its PU scaffold, where HA can be observed in PU/HA scaffolds.

**Figure 3 ijms-23-10333-f003:**
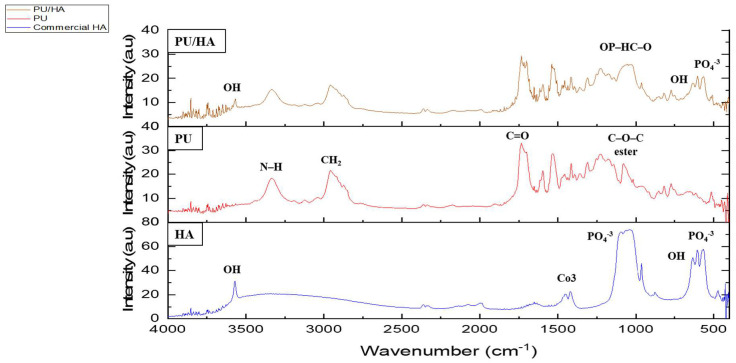
FTIR (PAS) spectrum comparing PU and PU/HA scaffolds’ main peaks.

**Figure 4 ijms-23-10333-f004:**
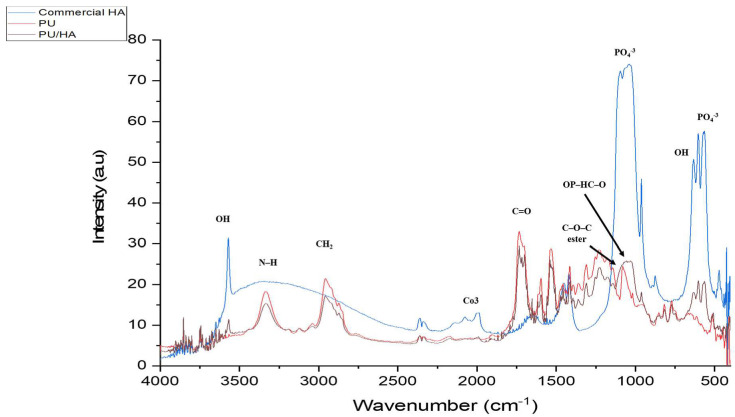
FTIR (PAS) spectrum overlap PU and PU/HA scaffolds related with main peaks.

**Figure 5 ijms-23-10333-f005:**
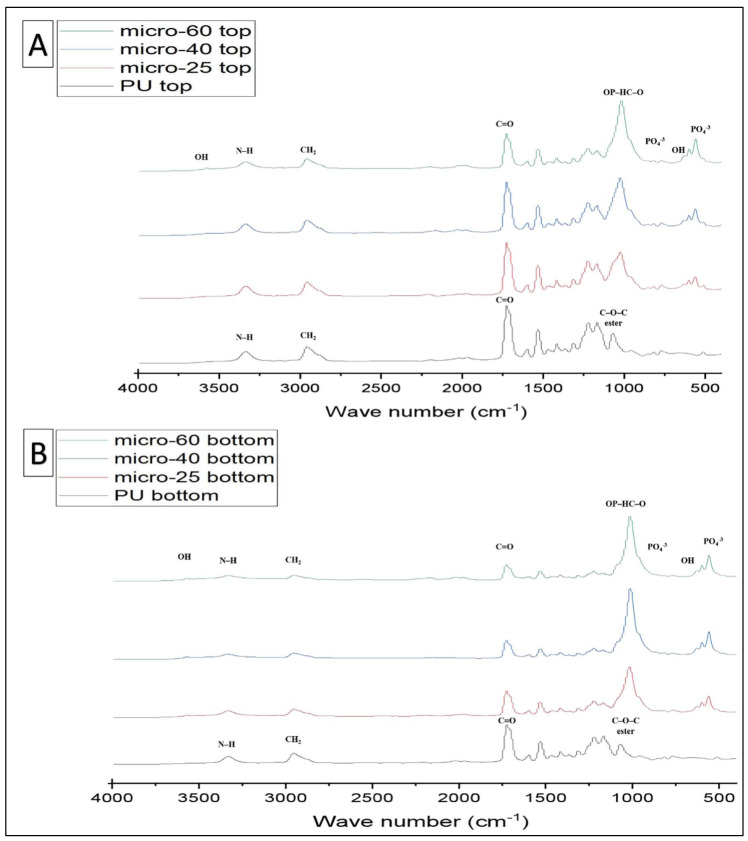
FTIR-ATR Spectra of different groups taking PU as a control (**A**) top surface of PU and PU/HA_micro_ scaffolds (**B**) bottom surface of PU and PU/HA_micro_ scaffolds.

**Figure 6 ijms-23-10333-f006:**
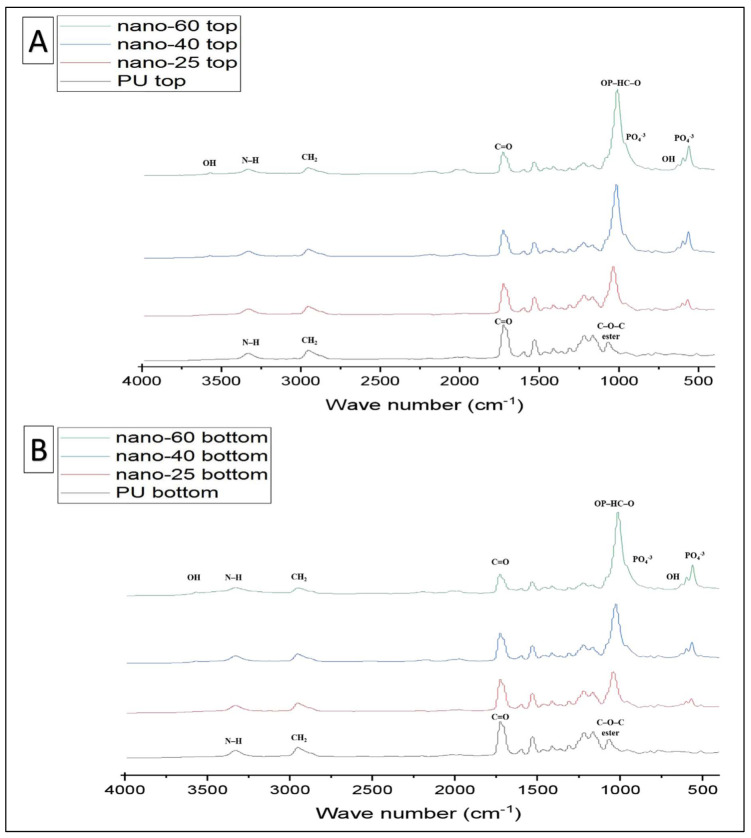
FTIR-ATR Spectra of different groups taking PU as a control (**A**) top surface of PU and PU/HA_nano_ scaffolds (**B**) bottom surface of PU and PU/HA_nano_ scaffolds.

**Figure 7 ijms-23-10333-f007:**
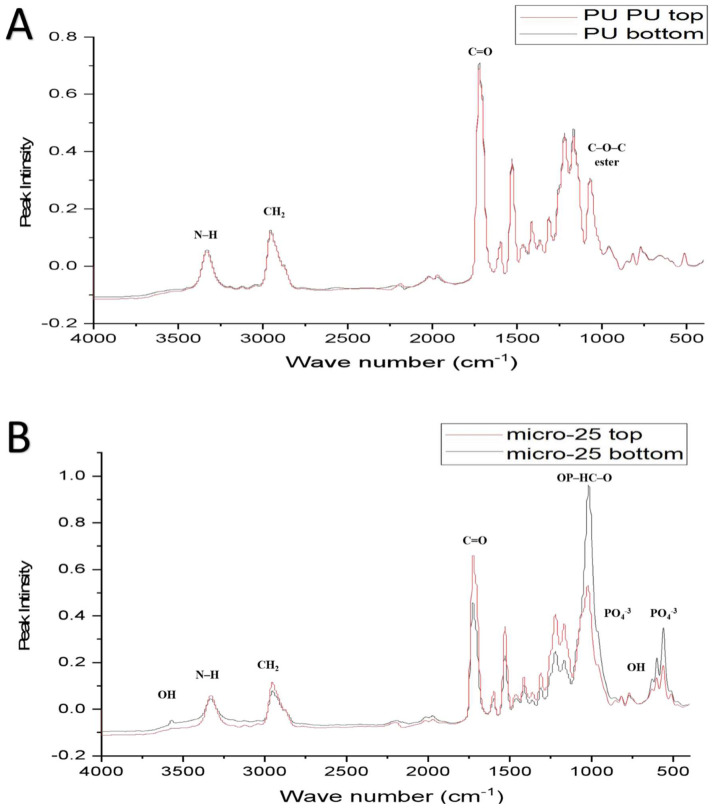
The combined FTIR-ATR spectra of top and bottom of each scaffold; (**A**) PU scaffold has no difference, (**B**–**D**) PU/HA_micro_ (25, 40 & 60%) scaffolds are different and (**E**–**G**) PU/HA_nano_ (25, 40 & 60%) scaffolds have almost no difference in peak intensity of OH, PO_4_ and CO_2_.

**Figure 8 ijms-23-10333-f008:**
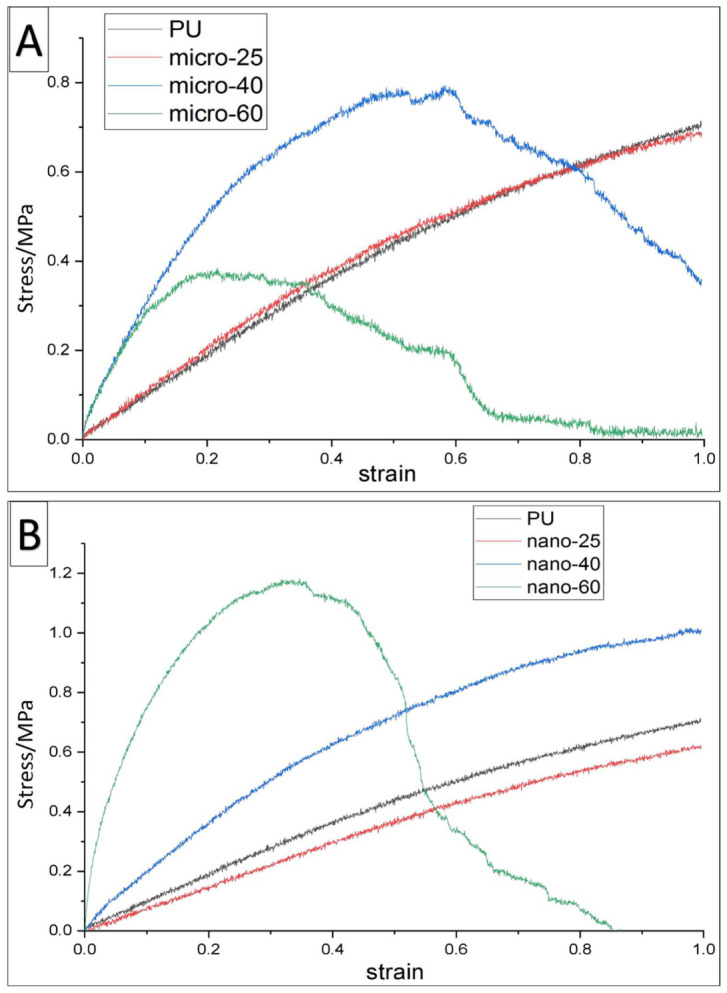
show stress/strain curves of PU and PU/HA scaffolds with 25%, 40% and 60% HA micro (**A**) and nano (**B**) particles respectively, note the different scale bars. Young’ s modulus and yield strength of scaffolds with statistics. Differences between two groups were distinguished as statistically significant if *p* ≤ 0.05 (*), *p* ≤ 0.01 (**) and *p* ≤ 0.0001 (****). (For the samples which are not broken in the tensile mechanical testing, the stress of the final point is considered as their yield strength.)

**Figure 9 ijms-23-10333-f009:**
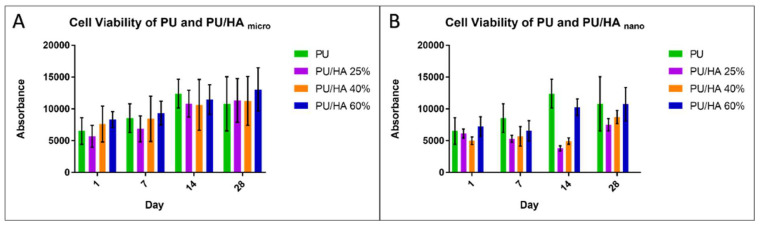
Alamar Blue cell viability; (**A**) for PU/HA_micro_ group & (**B**) for PU/HA_nano_ group. All scaffolds show good cell viability and biocompatibility.

**Figure 10 ijms-23-10333-f010:**
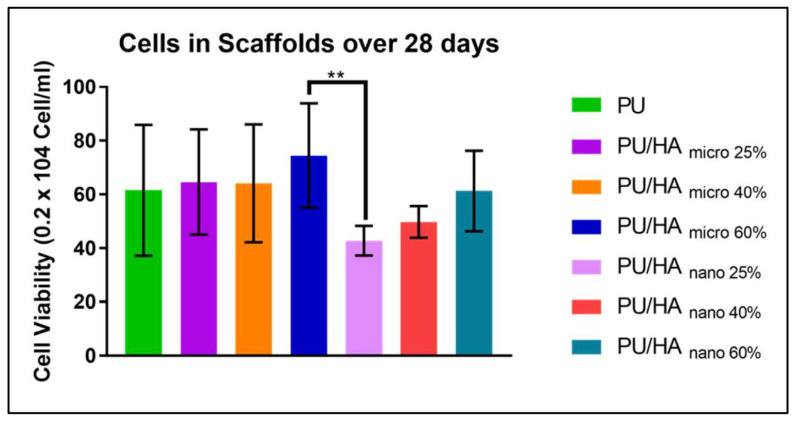
The overall number of metabolically active cells in each scaffold over 28 days. PU/HA_micro 60%_ was significantly higher than PU/HA_nano 25%._ Differences between two groups were distinguished as statistically significant if *p* ≤ 0.01 (**).

**Figure 11 ijms-23-10333-f011:**
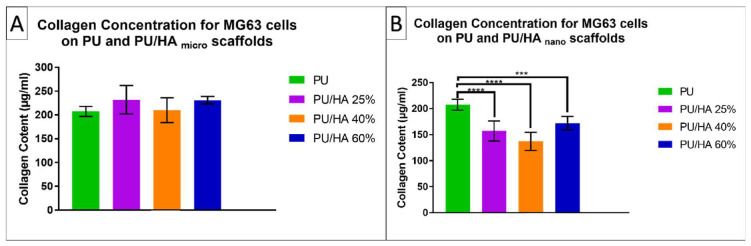
Collagen matrix production; (**A**) for PU/HA_micro_ group & (**B**) for PU/HA_nano_ group scaffolds on days 28 of culture. PU scaffold has higher than PU/HA _nano_ scaffolds significantly, where no significant difference with PU/HA micro. Differences between two groups were distinguished as statistically significant if *p* ≤ 0.001 (***) and *p* ≤ 0.0001 (****).

**Figure 12 ijms-23-10333-f012:**
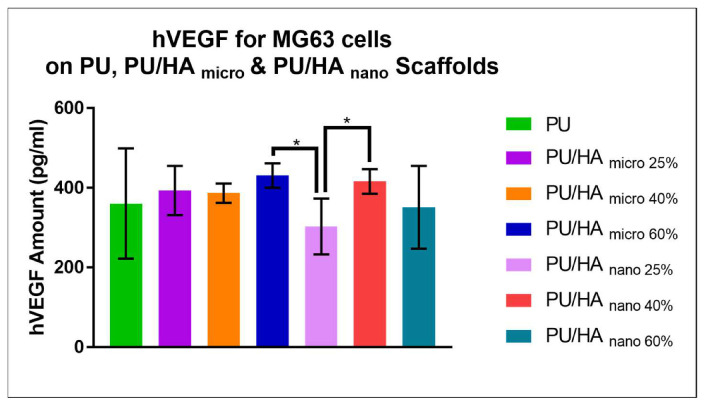
the VEGF protein amount was similar between PU, PU/HA_micro_ and PU/HA_nano_ scaffolds with no significant difference. Except on the PU/HA_nano 25%_, as it was significantly lower than PU/HA_micro 60%_ and PU/HA_nano 40%_. Differences between two groups were distinguished as statistically significant if *p* ≤ 0.05 (*).

**Figure 13 ijms-23-10333-f013:**
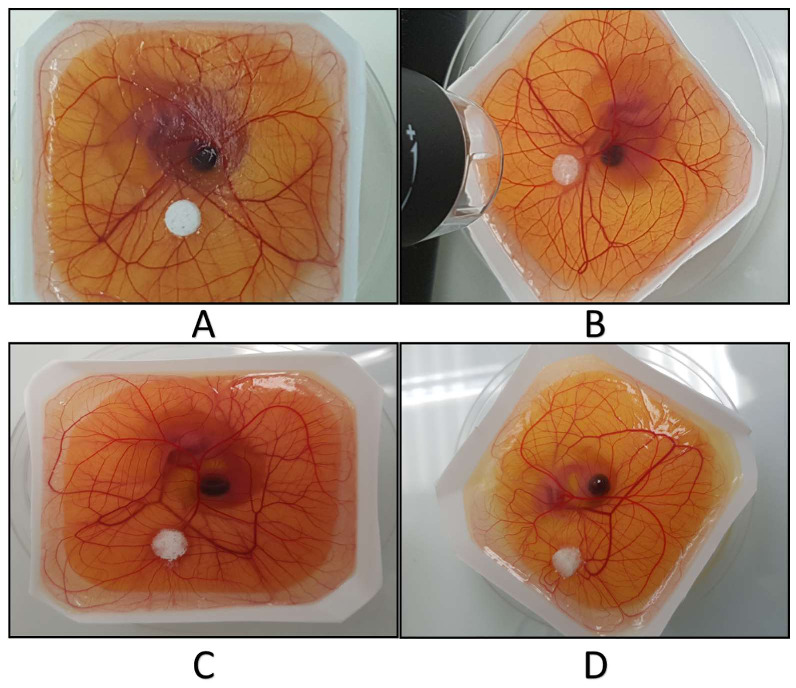
CAM Assay experiments, (**A**,**B**) PU scaffold of two different samples. (**C**,**D**) PU/HA scaffolds of two different samples.

**Figure 14 ijms-23-10333-f014:**
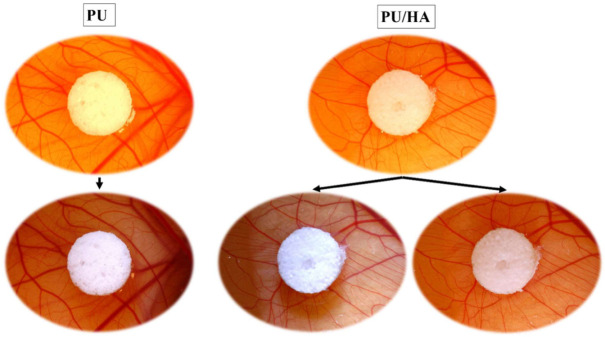
PU and PU/HA scaffold in the CAM essay to show comparison of new blood vessel occurrence with the scaffolds.

**Figure 15 ijms-23-10333-f015:**
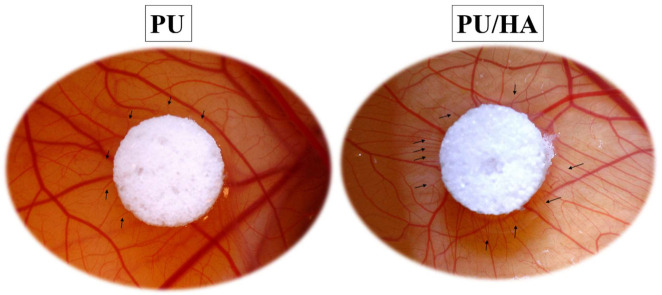
Comparison of new blood vessels forming around PU and PU/HA scaffolds. Scaffolds with HA has more arrows indicating blood vessels around the scaffolds compared to PU scaffolds.

**Figure 16 ijms-23-10333-f016:**
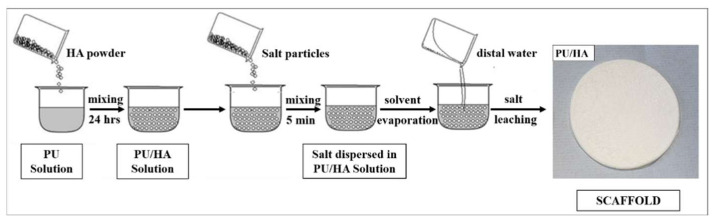
Diagram of Particles Leaching Method (PL) [62].

**Table 1 ijms-23-10333-t001:** The bands of PU and PU/HA scaffolds using FTIR [19,30,31].

Functional Groups	PU Scaffolds Wavenumbers cm^−1^	PU/HA Scaffolds Wavenumbers cm^−1^
O–H (stretching)		3570
Free N–H	3440, 3451	
Bonded N–H (stretching) (linkage of isocyanate and soft segment, known as urethane linkage)	3330	3330
C–H Aromatic region	3121, 3039, 3031	3121
N–H–C=O Linkage of OH, (HA) and NCO (PU) covalent bonding		3313
CH_2_ (Polyester), stretching vibration	2930–2800	
C–H band of (-CH_2_-) Present at a number of positions due to the presence of different (-CH_2_-) group in polymer chain	2958, 2885, 2796,	2930–2800
C–H band of (-CH_2_-), Chain extender	817, 772	817, 772
C–H aromatic regions	3122, 3101	
C–C aromatic ring breathings of (MDI)	1597, 1417, 517	
C=O hydrogen bonded secondary amide absorption bands	1750–1650, 1716 1734–1616	1710, 1716 1734–1616
C=O free (non-bonded), peak position changes due to the change in the soft segment	1734 1733–1728	1730 1734
N–H + C–N stretching	1540, 1232, 1224	1540, 1232, 1224
O–C–O + N–H (urethane linkage) interactions/bonding with phosphate groups, also causes a shift to the carbonyl absorption band, shifting it to a higher wavenumber	1716	1719
N–H + C–N, amide II	1540	1540
C–C, symmetric stretch, aromatic ring of MDI	1417	1417
N–H + C–N, symmetric stretch	1311, 1232	1311, 1232
C–O–C, ester linkage	1081, 1065	1081, 1085
C–H, aromatic ring breathings	1473, 1457,1019	
O–P–O, Phosphate		1074, 962, 603, 566
C–C, aromatic unsymmetric stretch	817, 517	817

**Table 2 ijms-23-10333-t002:** Components of PU and PU-HA solutions.

Name	DMF (wt%)	PU:HA (wt)	Micro-HA (%)	Nano-HA (%)
PU	100	4:0	0	0
Micro-25	100	3:1	25	0
Micro-40	100	2.4:1.6	40	0
Micro-60	100	1.6:2.4	60	0
Nano-25	100	3:1	0	25
Nano-40	100	2.4:1.6	0	40
Nano-60	100	1.6:2.4	0	60

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
