# Peer review of "Bioactive Composite for Orbital Floor Repair and Regeneration"

_ijms, 2022, doi:10.3390/ijms231810333_

Round 1

Reviewer 1 Report

this manuscript describing the enhancement to biocompatibility using micro and nano hydroxyapatite/polyurethane scaffolds is an interesting read and represents excellent research. the conclusions are supported by the data and the scientific approach is thorough, carefully designed, and obviously meticulously executed.

the only suggestions are:

-a proofread for grammar and consistency.

-a reworking of all the ftir and the stress/strain figures. most are blurry/too small or the labels are unreadable. it can be a challenge to display all this data without taking up a ton of space but the authors are encouraged to look around and see if there are better options. 

Author Response

We thank the reviewer for appreciating design and methodology of the research paper. 

Figures have been changes as suggested and combed through the paper again. 

FTIR and Stress/Strain figures have been changed as suggested, better quality images have been incorporated.

Figure 5 Page 8-9

Figure 6 Page 9-10

Figure 7 Page 10-14

Figure 8 Page 15-17

Labels have also been edited.

Reviewer 2 Report

Dear Authors,

The manuscript is very interesting and well written. The introduction is appropriate and explains the topic in which the paper fits very well. The length of this manuscript is appropriate and the subject is very interesting.

The materials and methods are in detail and reproducible in the laboratory without problem.

The results are statistically valid. The photographs and figures are very clear and very well described.

The discussion and conclusion are appropriate and well connected to the introduction and results.

The bibliography does not have many recent articles. You should include more recent articles in the references.

Author Response

References have been added , Please see reference numbers; 22, 23, 26 and 45

Reviewer 3 Report

Please add a short discussion of how ongoing bone remodeling may effect the long term outcome.

Author Response

We thank the reviewer for very positive feedback. 

As suggested the following has been added in response to the comment of bone remodelling and its long-term effect in relation to angiogenesis. 

Furthermore, angiogenesis is necessary for not only for bone formation and but also during healing/bone remodelling that shows the importance of angiogenesis for osteogenesis. Recently in our group, heparin has been  used to induce angiogenesis. It has been investigated via CAM assay to evaluate the potential attachment of physiologically available angiogenic growth factors to pro-angiogenic receptors by using heparin bonded chemically crosslinked chitosan poly-vinyl alcohol (PVA) hydrogels. Triethyl orthoformate (TEOF) crosslinked and heparin bonded hydrogels led to more blood vessel generation as compared to heparin-free control samples which promotes bone remodelling (58). In addition, direct mixing of heparin in chitosan-PVA-PCL hydrogels in the absence of any growth factors has been investigated for angiogenesis on chick embryo’s CAM tissues for wound healing application. It was found that heparin bonded chitosan-PVA-PCL hydrogels led to significantly more angiogenesis than the sole collagen control gels (59). Therefore, the functionalisation of scaffolds for its potential physiological binding activity can provide advantages in bone regeneration and remodelling.

Reviewer 4 Report

This manuscript describes the preparation of bioactive composites for orbital floor repair and regeneration. The article is well written, the Authors used a wide range  of metody for physicochemical characterization of obtained materials. In my opinion this article can be published i the present form. My only remark is that for 60 references only few articles was published in recent 5 years. The Authors should add and shortly discussed recent advances in this area.

Author Response

Recent references have been added and discussion elaborated. Please see reference numbers; 22, 23, 26 and 45.

Reviewer 5 Report

This is an extensive and excellent study conducted by the authors that demonstrate that hydroxyapatite enhances vascularization and its addition to the composite biomaterial comprising of polyurethane makes it very useful for orbital floor repair and regeneration. This discovery will modify the treatment methodologies for orbital floor regeneration, specifically in the maxillofacial area where regeneration or repair becomes quite difficult. This is a structured manuscript written with appropriate background. The results obtained support the hypothesis. The methods including statistical analysis are appropriately described. The authors have considered the recent developments in the area while discussing their findings. Overall, the manuscript is acceptable for publication in its current form.

Author Response

We thank the reviewer for positive feedback.